# Multigrid/Multiresolution Interpolation: Reducing Oversmoothing and Other Sampling Effects

Daniel Rodriguez-Perez [1,*,†] and Noela Sanchez-Carnero [2,3,†]

1 Departamento de Física Matemática y de Fluidos, Facultad de Ciencias, UNED, Avda. Esparta s/n, 28232 Las Rozas, Madrid, Spain
2 Centro para el Estudio de Sistemas Marinos (CESIMAR), CCT CONICET-CENPAT, Bv. Almirante Brown 2915, Puerto Madryn U9120ACD, Chubut, Argentina; noelas@gmail.com
3 Grupo de Ocenografia Fisica (GOFUVI), Facultade de Ciencias do Mar, Campus de Vigo, Lagoas-Marcosende, Illa de Toralla s/n, 36331 Vigo, Pontevedra, Spain
* Correspondence: daniel@dfmf.uned.es
† These authors contributed equally to this work.

**Abstract:** Traditional interpolation methods, such as IDW, kriging, radial basis functions, and regularized splines, are commonly used to generate digital elevation models (DEM). All of these methods have strong statistical and analytical foundations (such as the assumption of randomly distributed data points from a gaussian correlated stochastic surface); however, when data are acquired non-homogeneously (e.g., along transects) all of them show over/under-smoothing of the interpolated surface depending on local point density. As a result, actual information is lost in high point density areas (caused by over-smoothing) or artifacts appear around uneven density areas ("pimple" or "transect" effects). In this paper, we introduce a simple but robust multigrid/multiresolution interpolation (MMI) method which adapts to the spatial resolution available, being an exact interpolator where data exist and a smoothing generalizer where data are missing, but always fulfilling the statistical requirement that surface height mathematical expectation at the proper working resolution equals the mean height of the data at that same scale. The MMI is efficient enough to use *K*-fold cross-validation to estimate local errors. We also introduce a fractal extrapolation that simulates the elevation in data-depleted areas (rendering a visually realistic surface and also realistic error estimations). In this work, MMI is applied to reconstruct a real DEM, thus testing its accuracy and local error estimation capabilities under different sampling strategies (random points and transects). It is also applied to compute the bathymetry of Gulf of San Jorge (Argentina) from multisource data of different origins and sampling qualities. The results show visually realistic surfaces with estimated local validation errors that are within the bounds of direct DEM comparison, in the case of the simulation, and within the 10% of the bathymetric surface typical deviation in the real calculation.

**Keywords:** multiresolution interpolation; bathymetry; SRTM; Gulf of San Jorge; Patagonia; Argentina; Atlantic Ocean

## 1. Introduction

Digital elevation models (DEM) are important tools to study the Earth surface and model the processes taking place over it; hazard mapping, climate impact studies, geological and environmental modeling, atmospheric and marine flow simulations including tide prediction, are just a few of their current applications [1–4]. A grid DEM represents the continuous surface interpolated through (discrete) points where elevation has been measured and recorded, and is usually represented as an image whose pixels contain elevation data. High resolution DEMs (∼1 m) appearing in the late 1990s allowed geomorphological exploration with unprecedented detail, both by visual analysis of shaded DEM (e.g., that provides an easy inspection of features at various scales) [5] and through geomorphological indices quantified from the raster image [6,7]. Finding the best DEM

generalization (i.e., interpolation) for the scale of topographical features of interest is a key element for multiscale analysis of structural topographic features [5,8,9].

Assessing the accuracy of DEMs is a pending issue, especially for the submerged part of the Earth, where both density and distribution of acoustic bathymetric measurements [10] and spatial resolution (either of interpolation or of indirect gravimetric inversion) are limited. Furthermore, DEM quality is also affected by characteristics of the surface or terrain roughness, cell size or spatial resolution, and the chosen interpolation method (and decisions made about its parameters) [11,12].

Currently, there are different open access global DEMs of the emerged Earth with moderate resolution such as the shuttle radar telemetry model (SRTM, 1 arc second, approximately 30 m horizontal, and 16 m vertical resolution) [13], the ASTER global DEM (GDEM v3, 2.4 arc seconds, approximately 90 m horizontal, and 12 m vertical resolution) [14,15], the Japan Aerospace Exploration Agency (JAXA) AW3D high-resolution global digital surface model (5 m horizontal and 6.5 m vertical resolution) [16], and the ICESat GLAH14 (6 m horizontal and 15 cm vertical resolution) [17,18].

Mapping the submerged bottom of the seas and oceans has required more work. The best known example of open-source bathymetric DEM is the General Bathymetric Chart of the Oceans (GEBCO) [19,20]. Elaborating this DEM involves cleaning and harmonizing data sources and then interpolating them into a surface. Often, this is an iterative process as source data cleaning (and, sometimes, harmonization) cannot be done without an estimated DEM. The acquisition of acoustic data over large areas is very expensive (for a given spatial resolution, it grows with the square of the area), so crowdsourcing strategies are being used to build large databases of bathymetric information [21], being GEBCO one of the most successful ones in terms of integration from multiple sources.

Interpolation methods can be grossly grouped into deterministic, geostatistical and machine learning methods (see the reviews [22–24] for more details):

- Deterministic interpolation methods include nearest (natural) neighbour (NN) [25], inverse distance weighting (IDW) [26], or trend surface mapping (TS) [27]. These methods often work better with homogeneous distributions of data points. There are also models, as ANUDEM a.k.a. ArcGIS TOPO2GRID [28] that are designed to interpolate data along curves (e.g., isolines or river basins).
- Geostatistical interpolation is commonly known as kriging which estimates elevation using the best linear unbiased predictor, under the assumption of certain stationarity assumptions [29,30]. There are many variants that overcome some limitations about those statistical assumptions (such as indicator kriging), or improve prediction based on co-variables (co-kriging).
- Machine learning interpolation methods apply interpolation/classification methods to group "likewise" measurements thus enhancing their efficiency by using previous results. Despite the widespread use of machine learning, its use applied to spatial data is still a field of research; dealing with spatial heterogeneity and the problem of scale are areas in which these techniques can excel (see [31,32]). These methods are also showing their great potential when dealing with multi-source multi-quality data [33].

Interpolated DEMs often present "pimple" artifacts. These are typical of exact interpolation methods, where they appear around sampling points (quite common in IDW), but also appear in approximate (e.g., geostatistical) methods, and are usually removed by filtering the resulting DEM or by increasing the search window. This may cause; however, oversmoothing if the estimated correlation length is larger than the details available in particular areas with higher sampling density; this has been addressed by variance correction methods [34,35]. Another common artifact in DEMs are "transect" artifacts, very common in bathymetric DEMs, that appear where data density is higher (along transects) in contrast with the rest of the raster which is generalized. Some statistical resampling methods have also been devised to address this problem [36,37]. The non-uniform sampling of terrain data can be also caused by selection bias in topographic data (e.g., limited to easily accessible areas), leading to scarcely sampled areas compared with other highly sampled ones.

High accuracy surface modeling methods have been proposed which attempt to overcome this limitation by imposing differential geometry constraints that preserve the expected topographical continuity [38] or introducing pre-interpolated features (e.g., isolines) in the interpolation [36]. Of course, the alternative is to increase sampling effort in undersampled areas; however, this is not always feasible.

The spatial resolution required from a DEM depends largely on the focus of our study interest. For example, a continental DEM or an ocean-wide bathymetry do not require resolving details smaller than several kilometers. On the other hand, the study of coastal tidal dynamics or coastal geomorphometry, or lake or water dam bathymetry may require resolving details of tens of meters or even meters [4,39–42]. When dealing with large areas involving continental scale features data size grows rapidly making it almost impossible to efficiently estimate elevation at points where data are not available, hence techniques are required that are able to efficiently handle large data sets. This interest in multiple scales across large geographical areas has led naturally to multiscale algorithms, either to improve computation of traditional geostatistical interpolations [43], to get advantage of wavelet interpolation algorithms [44], to complete information (especially in bathymetries) by "superresolution" (techniques inherited from digital image inpainting) [45,46], to store and get access to scale-dependent information [47], to analyze scale-dependent geomorphological features [5,9], or even to extrapolate the topography to finer resolutions than available from the data in what is called geostatistical simulation [48,49].

Spatial interpolation methods, either multiscale or not, usually make assumptions about the sampling process (e.g., random independent point-wise sampling), surface statistical properties (e.g., gaussian height distribution, functional form of the variogram), neighborhood shape and extension (e.g., triangulation, look-up distance, look-up directions or quadrants), smoothness penalization or other parameters (curvature constraints, wavelet family, etc.). This makes the choice difficult in common working conditions, statistical assumptions difficult to test, and algorithm parameters difficult to adjust, being the "desirable visual aspect" the most used heuristic criterion in choosing the interpolation, and the software availability and computer memory and processing time the other criteria. The latter are very dependent on the number of points to be interpolated, which again calls for efficient multiresolution approaches.

The goal of this article is to describe multigrid/multiresolution interpolation (MMI) based on simple (if not simplistic) hypotheses about the data, and which is able to solve many of the problems other interpolation methods have, while being fast and extensible. For that we will first introduce a top-down multigrid/multiscale method which meets them while making the simplest hypotheses about the input data or about the interpolated surface (Section 2.1). Then we will show how to use it for surface extrapolation (assuming a self-affine multi-fractal terrain model, in Section 2.3), and cross-validation later used for data filtering and outlier detection (Section 2.4). We will apply this algorithm to two case studies in the area of the Gulf of San Jorge (in Argentina's Patagonia, described in Section 1): one based on synthetic data extracted from the SRTM DEM of the coastal area (Section 3.1), and another based on actual multi-source bathymetric data in order to compute the bathymetric surface of the Gulf (Section 3.2). We will discuss our proposal based on these case studies, and on the current bibliography (Section 4) and, finally, draw some conclusions.

## 2. Method

Mathematically speaking, interpolation means filling in the gaps of our information about a function based on the information we have about that function, especially but not limited to, the values that function takes at some known points. In what follows, we will construct a multigrid/multiresolution interpolation (MMI) method keeping in mind the geometrical relationships, and properties of exactness, regularity and smoothing, and statistical expectation of the methods described in the introduction. We will also focus on surface interpolation, i.e., interpolation of a real valued function $f$ defined on an interval

$I = [a, b] \times [b, c] \in \mathbb{R}^2$; without loss of generality, we will assume that interval to be $I = [0, 1] \times [0, 1]$.

### 2.1. Top-Down Multigrid/Multiresolution Algorithm

Although some interpolation methods aim at providing a grand final mathematical formula to approximate function $f$ at any point $x \in I$, often that formula is not used, but an iterative method estimates the value of $f$ at $x$ from its values $f(x_i)$ at the observation points $x_i \in I$. In addition, in practice, we are often interested in obtaining the average value of the function in some neighborhood $B \subset I$ of $x$, being the precise value at $x$ often inaccessible experimentally. Based on these two practical approximations, we formulate our multigrid method as follows:

1. Start with a partition of $I$ in $2^{n_0} \times 2^{n_0}$ intervals of the form

$$B_{ij} = \left[ i \times 2^{-(n+1)}, (i+1) \times 2^{-(n+1)} \right] \times \left[ j \times 2^{-(n+1)}, (j+1) \times 2^{-(n+1)} \right]$$

with $n = n_0 \in \mathbb{N}$ and $i, j = 0, 1, \ldots, 2^n - 1$. Thus the sidelength of each $B_{ij}$ is equal to $2^{-(n+1)}$, being the sidelength of $I$ equal to 1.

2. Chose those $B_{ij}$ such that for some $k$ there is some observation point $x_k \in B_{ij}$. Let us call $N_{ij}$ the number of those observation points inside $B_{ij}$ and estimate the average value of $f$ in $B_{ij}$ to be

$$\hat{f}(B_{ij}) = \langle f(x_k) \rangle_{x_k \in I_{ij}} = \frac{1}{N_{ij}} \sum_{x_k \in B_{ij}} f(x_k) \tag{1}$$

This means that our estimation of $f$ in $B_{ij}$ is the most likely one (maximum likelihood) given by the arithmetic mean of the $N_{ij}$ measured points inside $B_{ij}$.

3. Let us now focus on some $B_{ij}^*$ such that there is no $x_k \in B_{ij}^*$. Let us consider its neighbor intervals, of the form $B_{i \pm \{0,1\} \, j \pm \{0,1\}}$, such that the value of $\hat{f}$ could be computed in them; let us denote that set of neighbor intervals $\mathcal{N}_{ij}$. Then, we will interpolate

$$\hat{f}(B_{ij}^*) = \frac{\sum_{B \in \mathcal{N}_{ij}} w_B \hat{f}(B)}{\sum_{B \in \mathcal{N}_{ij}} w_B} \tag{2}$$

where the $w_B$ are weights assigned to intervals $B \in \mathcal{N}_{ij}$. The simplest weight assignment would be the number of points inside $B$, that is $w_{B_{kl}} = N_{kl}$, meaning that we take $B_{ij}^*$ as a part of the larger set $\bar{B}_{ij} = B_{ij}^* \cup \left( \cup_{B \in \mathcal{N}_{ij}} B \right)$ and then we estimate $\hat{f}$ as the average of $f$ over the points measure in that enlarged set $\bar{B}_{ij}$. Under this assumption, we can also interpolate the number of expected measurement points in $B_{ij}^*$ (e.g., after a new statistically independent measurement of the function) as

$$N_{ij}^* = \frac{\sum_{B \in \mathcal{N}_{ij}} w_B N_B}{\sum_{B \in \mathcal{N}_{ij}} w_B} \tag{3}$$

equating $N_{B_{ij}} = N_{ij}$ in subindices notation.

**Remark 1.** *For a partition of I with $n > n_0$, the expression "such that the value of $\hat{f}$ could be computed in them" will also include the rough estimation of $\hat{f}$ (and of $N_B^*$, $B \in \mathcal{N}_{ij}$) from the previous partition $n - 1$ given by (4) below.*

4. Now, we will refine the partition of $I$ by defining, for each $B_{ij}$ four subintervals (quadtree structure), $B_{ij,kl}'$ with $k, l = 0, 1$. If our partition of $I$ was made in $2^n \times 2^n$ intervals, then this one will be in $2^{n+1} \times 2^{n+1}$ intervals of the form

$$B'_{ij,kl} = \left[ (2i+k) \times 2^{-(n+2)}, (2i+k+1) \times 2^{-(n+2)} \right] \times \left[ (2j+l) \times 2^{-(n+2)}, (2j+l+1) \times 2^{-(n+2)} \right]$$

and assign to each of these subintervals the following values of $\hat{f}$ and $N'_{ij,kl}$ (until a better approximation is made)

$$\hat{f}(B'_{ij,kl}) = \hat{f}(B_{ij}) \tag{4}$$

$$N'_{ij,kl} = \frac{1}{4} N_{ij}$$

5. At this point, we have for the partition of $I$ in $2^{n+1} \times 2^{n+1}$ intervals a rough estimation of $\hat{f}$, $N'_{ij,kl}$ in each of its subintervals. Then, we can relabel those $B'_{ij,kl}$ subintervals applying the substitution $(ij, kl) \to (2i+k, 2j+l)$ and go back to step 2 to calculate an improved interpolation on a new $n+1 \to n$ partition in new updated intervals $B_{ij}$ of side-length $2^{-n}$.

The multigrid quadtree refinement structure of the algorithm makes it to reach a spatial resolution of $r$ (i.e., $r$ is the sidelength of any of the $B_{ij}$ intervals in the last iteration) in $-\log_2(r) - n_0 + 1$ iterations of the previous 5 steps. We only run through the scales in one direction, top-down, hence the title of this section.

*2.2. Some Properties of the Algorithm*

**Exactness:** The method is an exact interpolator meaning that, for any partition of $I$ in $2^n \times 2^n$ subintervals, the interpolated $\hat{f}(B_{ij})$ is the mean of observed values of $f$ at points within $B_{ij} \subset I$, in particular for $B_{ij}$ containing one single point (that is the usual meaning of exact interpolation method).

**Smoothing:** Smoothing of the surface is done during the down-scaling process, applying a nearest neighbors weighted averaging (2) and (3). The neighborhood can be extended to only first-neighbors or to second-neighbors or can be weighted unevenly (e.g., assigning 0.614 weight to second neighbors, assuming octogonal symmetry). In order to get smoother surfaces, the application of Equation (1) can be stopped at some resolution $n_s$, applying from there on only the generalization operation; then, the method will not be exact at the highest resolution (i.e., pointwise).

**Statistical expectation:** At every resolution level $n$, pixels containing data points are asigned the average value of elevation, which is an unbiased estimator of the mean. However, pixels not containing data points are estimated from their surrounding pixels either at that resolution, $n$, if they contain data points, or at the previous resolution, $n-1$, if they do not. Equations (2) and (3), when used to estimate $\hat{f}(B^*_{ij})$ and $N^*_{ij}$ using as $w_B$ the $N_{ij}$ known up to that level, operate as unbiased estimators acting on unbiased estimations, and then will provide the unbiased expected value of $f(B_{ij})$ when averaged over all possible data samplings. As for the case of ordinary kriging, the underlying hypothesis is that $f$ is "locally constant", hence the neighborhood averaging.

**Sensitivity to outliers:** As long as the method is based on data averages (or estimated averages), outliers will have their effect on the results. They cannot be safely removed unless strong statistical assumptions (for instance, based on asymptotic standard error of the mean) are made scale-wide, because the same error correction should be applied at all scales. This will be assessed using $K$-fold cross-validation (see Section 2.4 below).

*2.3. Fractal Extrapolation*

Geological surfaces, and particularly bathymetric surfaces, are known to evolve through some of these scale-independent transformations and have often been charac-

terized as self-affine fractals [50] or multifractals [51–54] whose Hurst exponent or multi-fractality spectrum can be related to their geophysical evolution [6,53].

The well known "middle point displacement" method [55] has been used to construct visually realistic ladscape surfaces, and it applies a simple rule to succesively refine a triangulated surface (with some degree of randomness). Although there are variants to this method (among others, to generate multifractal surfaces [56]), the key idea is to make a refinement of the triangulated surface by inserting a new point inside each of its faces (e.g., at the center of the triangles) and assigning to it a height equal to some average of the previous triangle vertices heights plus a randomly distributed zero-mean displacement with variance $\sigma^2$ proportional to $L^{2H}$, being $L$ the side-length of the triangle. The new points, once included in the triangulation, multiply the number of triangles by 3, and the new triangulated surface is transformed by applying the same rule until the required spatial resolution (defined by the triangle side-length $L$) is achieved.

Given the similarities of this "middle point displacement" construction with our interpolation method, we will adopt it to modify Equation (2) in order to allow for a fractal simulation (or extrapolation) of $\hat{f}$ in those intervals $B_{ij}^*$ without actual measurements $x_k$. So we will just estimate

$$\hat{f}(B_{ij}^*) = \frac{\sum_{B \in \mathcal{N}_{ij}} w_B \hat{f}(B)}{\sum_{B \in \mathcal{N}_{ij}} w_B} + \frac{1}{\sqrt{12}} s_n \times \eta$$

where $\eta$ is a uniformly distributed random variable in $[-1, 1]$ and $s_n$ is the roughness of the surface (typical deviation) at the scale $L = 2^{-(n+1)}$, given by

$$s_n = \sigma_r \times (L/r)^H$$

where $r$ is the reference resolution (usually, the final interpolated map resolution), $\sigma_r$ is the estimated roughness at that resolution $r$ (i.e., the root mean square difference between surface heights measured at that resolution) and $H$ is the Hurst exponent.

Usually, $H$ will not be known beforehand, so it can be estimated:

**Globally:** from the globally mean roughness at the smallest scale (one pixel of the final interpolated map) computed from neighbor height differences $\Delta f$ between intervals containing observation points. If there are such $K$ pairs of neighboring intervals, then $\sigma_r^2 = s_N^2 = \frac{1}{K} \sum_{k=1}^{K} (\Delta f)^2$. The value of $H$ is estimated from the previous resolution roughness, $s_{n-1}$ which is already known: $H = \log(s_{n-1}/s_N) / \log(2L/r)$. Going global, maximizes the number $K$, thus the estimation is improved, however local roughness could vary from part to part of the domain.

**Locally:** in this strategy a value is estimated for $\sigma_r^2$ in each interval, using only the neighbor height differences $\Delta f$ of observation points within that $n$-th resolution interval (of size $L$). However, whenever there are no pairs of neighboring points within that interval, $\sigma_r^2$ is estimated from the previous resolution (of size $2L$) by the same interpolation method used to estimate $\hat{f}$. This implies that not only $\hat{f}(B_{ij})$ has to be interpolated, but also $\hat{\sigma}_r(B_{ij})$ using the same algorithm.

We will use the local approach in this article.

**Remark 2.** *Notice that the global estimation of H would play the role of the covariance structure estimation used in ordinary kriging, assuming a power law semivariogram model for the entire area, i.e., assuming a stationary covariance structure. The local approximation would allow for a non-stationary process similar to universal kriging, and also results in multifractal structures. The main difference here is that, as long as possible, the fractal structure is computed as close to the actual scale as possible from measured data, only applying the simulation where necessary, i.e., on intervals with no data for estimation.*

*2.4. Surface Validation and Error Estimation*

We would like to know how accurate the surface estimation is given a random sample of measurement points $(x_i, f(x_i))$. The common method to assess goodness of fit is validation, that is, using a part of the points not used to fit the function $f$ to compute the distance between the estimated values of $\hat{f}$ at those points and those actually measured values. However, this method only provides a pointwise (at each $x_i$) or a global (e.g., the mean square error) estimation of error. A bootstrap cross-validation, on the other limit, would repeat the interpolation a large number of times $K$ using each time an independent random sample (extracted "with repetition") of meaurement points, and then estimating the local interpolation error from the distribution of interpolation replicas $\{\hat{f}^{(k)}\}_{k=1...K}$.

In this article, we use a more modest and realizable estimation process, based on $K$-fold cross-validation. Interpolation will be repeated $K$ times, leaving each time $1/K$-th of the data out. Then, instead of only testing the accuracy of the interpolation with respect to that $1/K$-th of the data, we will estimate the local interpolation standard error $\Delta\hat{f}_{CV}(x)$ from the set of $K$ interpolation replicas $\{\hat{f}^{(k)}\}_{k=1...K}$ as:

$$\Delta\hat{f}^2_{CV}(x) = \sum_{p=1}^{K} \left[ \hat{f}^{(p)}(x) - \hat{f}_{CV}(x) \right]^2$$

where

$$\hat{f}_{CV}(x) = \frac{1}{K} \sum_{q=1}^{K} \hat{f}^{(q)}(x)$$

is the mean cross-validation surface.

**Remark 3.** *Apart from the obvious problem of computing a large number of interpolations posed by bootstrap, the condition of independent random samples poses a problem when measurement data are inherently correlated, as is the case with sampling transects. To address the problem of spatial correlation of points along a transect, we will adopt an "object oriented" K-fold partition of the data. We will subset each transect in smaller sub-transects of equal length (25 km was a practical choice for the case studies below), randomly assigning each of them to one of the K partitions of the data. We will use K = 10, which is a common choice in the literature [57].*

## 3. Case Studies

In this section we will apply our interpolation method to reconstruct and assess the quality of two surfaces interpolated from sampled data. First, we will sample data from an area of the SRTM digital elevation model, and test the accuracy of our interpolation both from the sampled data (using the $K$-fold error estimation) and from comparison with the actual model. Then, we will use bathymetric measurements acquired over an area equivalent in size, and compute the accuracy of our interpolation from those sampled data; in this case, we do not have a more accurate (i.e., computed from more extensive data) bathymetric model than our result, hence the interest of the first one.

Our study cases are located in the Gulf of San Jorge (GSJ) and its adjacent coastal area. The GSJ is is the largest gulf of the Argentinian Patagonian shelf, with an extension of 39,340 km$^2$ and a mouth of nearly 250 km, located between 45° S (Cape Dos Bahías) and 47° S (Cape Tres Puntas) (Figure 1). This gulf is a semi-open basin mainly covered by silt with coarse granulometric fractions to the north and south ends of the gulf [58,59], that reaches about 100 m of depth in its center, and having in its mouth depths ranging from about 90 m on the north and center, to 50–60 m on the south end, where the basin is demarcated from the adjacent shelf by a pronounced sill. The tidal regime in the GSJ is semidiurnal, with tidal amplitudes ranging between 3–5 m [60,61].

The continental vicinity of the GSJ forms part of the hydrocarbon-producing GSJ basin surrounded by the North Patagonian Massif (north), Deseado Massif (south), and the Andes (west) [62]. These massifs appear in the GSJ as Jurassic rhyolitic volcanic rock outcrops,

the larger one located in the northeast (close to Cape Dos Bahías). The GSJ basin plateau is mainly covered by Eocene-Miocene sedimentary rocks of the Sarmiento and Patagonia Formations [63], as well as Quaternary fluvio-glacial deposits ("Rodados Patagónicos"; [64]). This plateau reaches the coast as cliffs or gravel/sand beach-ridges [60].

The GSJ is a very interesting and complex case of management since several interests coexist in it [65]. On the one hand, GSJ is one of the most relevant areas of Argentina coast in terms of biodiversity and productivity with relevant areas for marine conservation because of the presence of reproductive aggregations and foraging grounds of many marine birds and mammals. Moreover, it houses major fisheries targeting valuable shrimp, hake, scallops and king crab stocks [66,67]. On the other hand, its hydrocarbon-producing geology makes it ground of offshore oil platforms [62]. Since each of these processes and activities (oceanographic, fisheries, oil platforms, etc.) extend beyond the limits of the Gulf, we have included in our study the adjacent areas, limited to the north by 44°20′ S (Cabo Raso), south by 48°05′ S (Punta Buque) and east meridian 64° W (Figure 1).

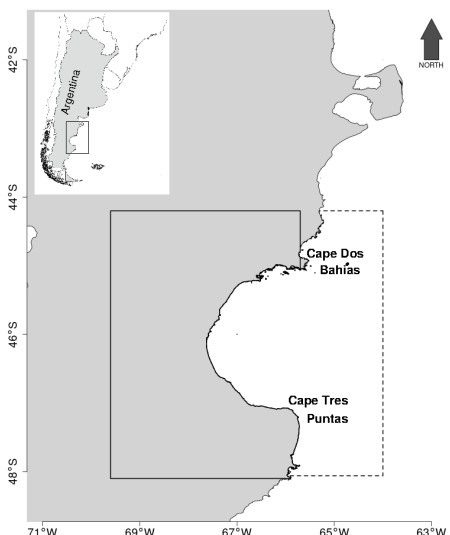

**Figure 1.** The area of Gulf of San Jorge with the delimitation of the land and ocean regions where the MMI algorithm has been tested.

### 3.1. SRTM Digital Elevation Model Sample Reconstruction

We selected the area between 69°6 and 65°7 W and between 48°1 and 44°2 S shown in Figure 1 (solid line rectangle) for our experiments. The SRTM30 tiles corresponding to this area were merged, resampled and reprojected onto a 90 m UTM grid (zone 19 S); the area includes a total surface of 84,400 km$^2$ in the emerged zone. Data samples were extracted using two different sampling strategies:

1. random point subsampling;
2. transect subsampling with 25 km long straight parallel transects.

Sampling density, that is the fraction of land points of the grid included in these samples was set to $p = 2^{-n}$, with $n = 4, 5, 6, 7, 8$ (that is, from $p \simeq 0.004$ to $0.063$). From those samples, a digital elevation model was interpolated with and without fractal extrapolation. For every sampling strategy and density, the average interpolation bias

$$\Delta \hat{f} = \langle \hat{f}_{\text{CV}} - f \rangle$$

root mean square error

$$\Delta \hat{f}_{\text{rms}} = \sqrt{\langle (\Delta \hat{f})^2 \rangle}$$

the 50% and 90% interquantile ranges of $\Delta\hat{f}$, denoted $\mathrm{IQ}_{50\%}\Delta\hat{f}$ and $\mathrm{IQ}_{90\%}\Delta\hat{f}$, and the correlation coefficient between $\hat{f}_{\mathrm{CV}}$ and $f$, $\mathrm{cor}(\hat{f}_{\mathrm{CV}}, f)$, were computed by direct comparison of the estimated $\hat{f}$ with the full SRTM data $f$. The $K$-fold cross-validation mean square errors

$$\Delta\hat{f}_{\mathrm{CVrms}} = \sqrt{\langle \Delta\hat{f}_{\mathrm{CV}}^2 \rangle}$$

are also included in Tables 1 and 2. The $K$-fold cross-validation estimated standard error $\Delta\hat{f}_{\mathrm{CV}}(x)$, as well as the standard error map of the interpolated surface, from which table values were computed, are shown in Figure 2. The sampling density of the highlighted column, $p = 2^{-6} \simeq 0.0156$, is the closest one to the sampling density of our case in Section 3.2, $p = 0.0181$.

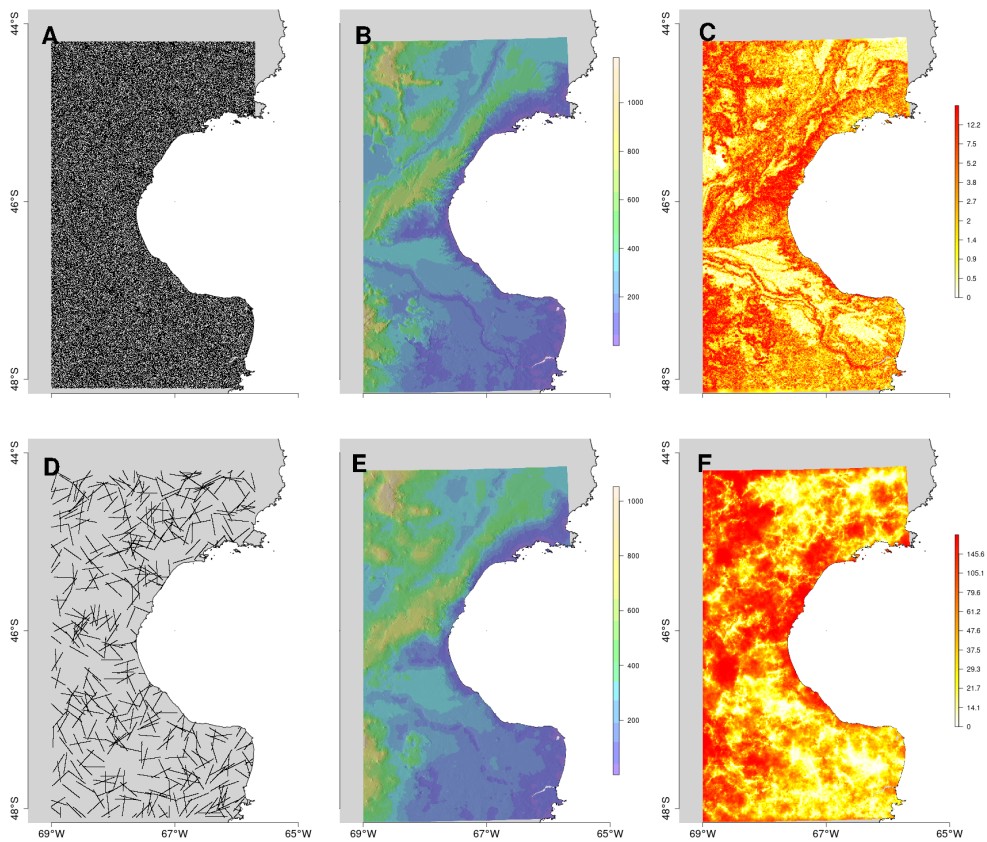

**Figure 2.** (**A**) Random points used to sample SRTM90 with with $p = 0.0156$; (**B**) Interpolated DEM ($\hat{f}_{\mathrm{CV}}$) from point samples; (**C**) $K$-fold cross-validation standard error $\Delta\hat{f}_{\mathrm{CV}}$. (**D**–**F**) Same meaning, respectively, but using transect sampling.

**Table 1.** *K*-fold and other statistics of interpolated DEM using simulated data extracted from SRTM at randomly distributed points ($p$ denotes point density per pixel). MMI was applied without and with fractal extrapolation. The SRTM90 column contains an assessment of SRTM resampling error based on the original 30 m resolution SRTM (using *K*-fold cross-validation) for comparison. Values are in meters.

| | Simple Interpolation | | | | | Fractal Extrapolation | | | | | SRTM90 |
|---|---|---|---|---|---|---|---|---|---|---|---|
| $p =$ | $2^{-8}$ | $2^{-7}$ | $2^{-6}$ | $2^{-5}$ | $2^{-4}$ | $2^{-8}$ | $2^{-7}$ | $2^{-6}$ | $2^{-5}$ | $2^{-4}$ | |
| $\bar{z}$ | 308.6 | 308.7 | 308.7 | 308.6 | 308.6 | 308.6 | 308.7 | 308.6 | 308.6 | 308.6 | 308.4 |
| $\sigma_z$ | 184.2 | 185.1 | 185.6 | 185.9 | 186.2 | 184.2 | 185.1 | 185.7 | 186.0 | 186.2 | 187.2 |
| $\Delta\hat{f}_{\text{CVrms}}$ | 12.55 | 10.15 | 8.23 | 6.65 | 5.40 | 19.33 | 16.47 | 15.18 | 14.58 | 15.75 | 2.50 |
| $\Delta\hat{f}$ | 0.044 | 0.152 | 0.087 | 0.045 | 0.065 | 0.052 | 0.182 | 0.115 | 0.023 | 0.088 | −0.007 |
| $\Delta\hat{f}_{\text{rms}}$ | 20.23 | 16.22 | 13.11 | 10.54 | 8.48 | 20.85 | 16.80 | 13.78 | 11.44 | 9.90 | 0.85 |
| $\text{IQ}_{50\%}\Delta\hat{f}$ | 8.00 | 6.53 | 5.10 | 4.01 | 3.13 | 12.72 | 10.34 | 8.58 | 7.32 | 7.02 | 1.25 |
| $\text{IQ}_{90\%}\Delta\hat{f}$ | 26.45 | 21.28 | 17.24 | 13.86 | 11.14 | 37.88 | 32.33 | 28.95 | 26.51 | 27.35 | 5.10 |
| $\text{cor}(\hat{f}_{\text{CV}}, f)$ | 0.9945 | 0.9965 | 0.9975 | 0.9985 | 0.9990 | 0.9944 | 0.9964 | 0.9973 | 0.9984 | 0.9989 | 1.000 |

**Table 2.** *K*-fold and other statistics of interpolated DEM using simulated data extracted from SRTM along random 25 km transects; $p$ denotes the fraction of the raster sampled by the transects. MMI was applied without and with fractal extrapolation. The SRTM90 column shows SRTM resampling error (see Table 1). Values are in meters.

| | Simple Interpolation | | | | | Fractal Extrapolation | | | | | SRTM90 |
|---|---|---|---|---|---|---|---|---|---|---|---|
| $p =$ | $2^{-8}$ | $2^{-7}$ | $2^{-6}$ | $2^{-5}$ | $2^{-4}$ | $2^{-8}$ | $2^{-7}$ | $2^{-6}$ | $2^{-5}$ | $2^{-4}$ | |
| $\bar{z}$ | 298.5 | 311.0 | 310.9 | 307.9 | 307.8 | 299.4 | 311.3 | 310.9 | 307.9 | 307.8 | 308.4 |
| $\sigma_z$ | 140.0 | 170.6 | 180.7 | 178.4 | 182.6 | 139.7 | 170.2 | 180.8 | 178.2 | 182.5 | 187.2 |
| $\Delta\hat{f}_{\text{CVrms}}$ | 73.33 | 53.39 | 52.70 | 31.49 | 23.98 | 136.55 | 109.22 | 88.23 | 61.90 | 44.72 | 2.50 |
| $\Delta\hat{f}$ | −9.992 | 2.467 | 2.355 | −0.616 | −0.698 | −9.168 | 2.775 | 2.403 | −0.684 | −0.708 | −0.007 |
| $\text{Var}\Delta\hat{f}$ | 105.56 | 79.76 | 55.92 | 40.38 | 28.05 | 111.59 | 85.59 | 60.45 | 43.97 | 30.68 | 0.85 |
| $\text{IQ}_{50\%}\Delta\hat{f}$ | 49.02 | 42.17 | 32.34 | 20.42 | 14.99 | 89.00 | 78.49 | 66.26 | 44.98 | 31.60 | 1.25 |
| $\text{IQ}_{90\%}\Delta\hat{f}$ | 153.37 | 108.89 | 106.08 | 66.91 | 50.70 | 247.59 | 196.91 | 172.97 | 126.35 | 92.25 | 5.10 |
| $\text{cor}(\hat{f}_{\text{CV}}, f)$ | 0.8430 | 0.9113 | 0.9589 | 0.9787 | 0.9899 | 0.8398 | 0.9082 | 0.9582 | 0.9780 | 0.9895 | 1.000 |

*3.2. Gulf of San Jorge Bathymetry Interpolation*

Now our area is comprehended between 67°7 and 64°0 W and between 48°1 and 44°2 S as shown in Figure 1 (dashed line rectangle), enclosing a marine area of 85,600 km². We used a number of data sources with different spatial sampling strategies (along transects and pointwise), densities, depth reference levels, etc.:

1.  Acoustic data from single and split-beam echosounders (SBES): This type of data is distributed in transects, within which there is a very high density of sounding points (depending on the vessel speed and the ping rate, but not greater than one sounding point every ten meters). In addition, the vertical resolution, although dependent on the working frequency, is usually less than 50 cm. In our study case we have several sources of this bathymetric information:

    *   The bathymetric data repository published by the National Institute for Fisheries Research and Development (INIDEP) of Argentina, which regularly conducts stock assessment surveys. This repository has a horizontal resolution of one sounding point every 5 m (see details in [68]). In our study area, there were 85085 sounding points, with depths between 11.5 and 123.1 m. These data are distributed in transects located mainly in the northern and southern areas of the GSJ, with less density in the central area.
    *   Data from oceanographic campaigns collected in the framework of research project PICT 2016-0218, from the analysis of oceanographic and fishing campaigns carried out by different Argentine intitutions. This database consisted of

> 147, 755 bathymetric points, with depths between 4.2 and 146.7 m. These data are distributed throughout the study area in transects with a mostly NW-SE orientation.
- Data from coastal campaigns. There were 4281 bathymetric points, with depth values between −2 (negative means above low-tide level, that is, the intertidal area) and 71.6 m, all of them acquired with portable echosounders from small vessels. These data are in areas very close to the coast, in the north of the GSJ.

Considering the tidal amplitude ranges in the GSJ, in order to refer all measured depths to a reference low-tide level, a tide correction was applied using the open OSU Tide Prediction Software (OTPS, available from https://www.tpxo.net/otps; access date 17 June 2022) [69].

2. Acoustic data from Multibeam (MBES) and Interferometric Sidescan Sonar (ISSS), which are acoustic sounders that, unlike SBES, provide wide swath coverage, at very high vertical and horizontal resolutions (up to a few centimeters). For our study area, these data come from three acoustic surveys in coastal areas (north of the GSJ), two with MBES and one with ISSS. For this work, the bathymetric surfaces were subsampled onto a 50 m grid. In total, 11, 305 bathymetric points were included, with depth values between 5.2 and 121.3 m.

3. Data from nautical charts: the basic source of bathymetric information are always nautical charts, in this case developed and maintained by the Naval Hydrography Services (Servicio de Hidrografía Naval) of Argentina. For our study area, data from six nautical charts were used; one of these charts, covered the entire area, while the other five cover smaller coastal areas, located to the north and west of the gulf, with higher detail. In total, 3522 bathymetric points were used, with depths between 0.3 and 119 m deep.

4. Data from the citicen-science project "Observadores a bordo" (on-board observers, POBCh). Most of the GSJ waters are under the jurisdiction of the province of Chubut, whose Fisheries Secretariat developed the program POBCh for years to control fisheries. In this program, along with fishing data, depth data were taken at those places where fishing sets were made (along with information of date and time). After this database depuration, we used 38,249 bathymetric points in our study area, with depths between 2.8 and 123 m and distributed throughout the entire GSJ except for the SW quadrant, which is under the jurisdiction of another province. Depth data were also corrected using OTPS based on observers annotated coordinates and local time.

5. Coastline. The 0 m isoline of the SRTM30 model was used as the union limit between the emerged and submerged areas. Points were generated along this line, that also includes islands, separated by 20–30 m (a second of arc, corresponding to the SRTM resolution) and with a depth value of 0 m. For the study area, 59,128 points were included from Santa Elena Bay, to the north, to Punta Buque. Coastline is used as a boundary condition and thus not included in the cross-validation process (i.e., it is always included in the interpolation) [36].

In order to harmonize the data, they were subsampled to take one point every 50 m along every transect (to reduce importance bias caused by larger sounding densities) and projected onto a 90 m UTM grid (zone 20 S). Whenever a new data source was projected onto this grid, its depth measurements were corrected to agree on average with the already projected data sources; as a reference, nautic charts were added second, just after the coast line data. The total number of data points within the study area was 248,443.

**Remark 4.** *Although in some sense this variety of bathymetric sources can be seen as crowdsourced data, all of the data sets were acquired in the context of scientific research programs, and had been previously curated and applied quality tests to remove erroneous data. For example, SBES acoustic data transects were tested for false bottom detections and missing echoes. Similarly, POBCh were checked for the existence of points far off their neighbor depths (usually erroneous manual annotations), and those points were removed from the dataset.*

Outlier Detection

Input data contained a number of points that cross-validation revealed as far-off the mean interpolated surface, sensibly farther than the local standard error $\Delta \hat{f}_{\text{CV}}(\boldsymbol{x}_i)$. To detect them and remove them from the input data we applied the algorithm known as Tukey fences [70] to measurement errors $\hat{f}_{\text{CV}}(\boldsymbol{x}_i) - f(\boldsymbol{x}_i)$. The algorithm consists in calculating the interquartile interval of all these measurement errors and removing those points departing from either interval bound more than $k_{\text{Tuck}}$ times its length. That is, only observation points such that

$$Q_{25\%}\Delta \hat{f}_{\text{CV}} - k_{\text{Tuck}} \times IQ_{50\%}\Delta \hat{f}_{\text{CV}} \quad < \quad \hat{f}_{\text{CV}}(\boldsymbol{x}_i) - f(\boldsymbol{x}_i) \quad < \quad Q_{75\%}\Delta \hat{f}_{\text{CV}} + k_{\text{Tuck}} \times IQ_{50\%}\Delta \hat{f}_{\text{CV}} \tag{5}$$

are kept. According to [70] a value $k_{\text{Tuck}} = 1.5$ does detect outliers, and a value of $k_{\text{Tuck}} = 3$ detects "far off" points; we have used $k_{\text{Tuck}} = 2$ here. We also removed points where $\Delta \hat{f}_{\text{CV}}(\boldsymbol{x}_i)/\hat{f}_{\text{CV}}(\boldsymbol{x}_i) > 0.5$, that is, the cross-validation relative standard error was above 50%; those points did not clearly contribute any information to the interpolation. In total 18,080 points were removed based on these criteria from interpolation in the study area.

After this, another interpolation was carried out again giving the results summarized in Figure 3 and Table 3.

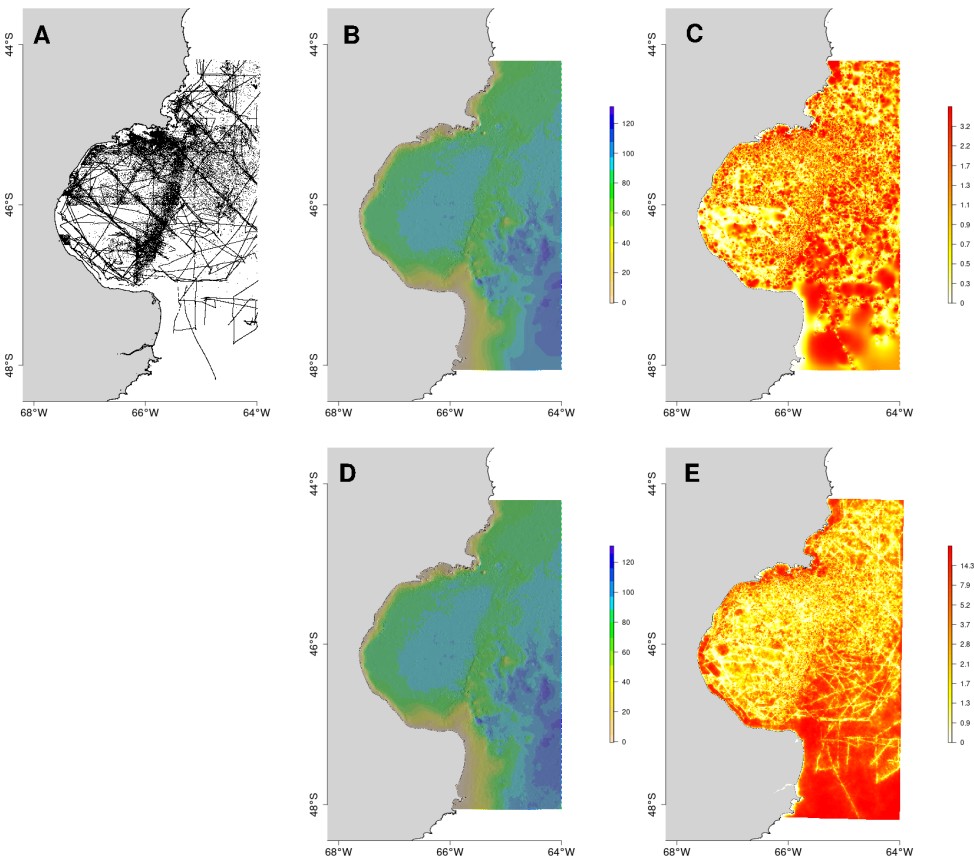

**Figure 3.** (**A**) Bathymetric acoustic sounding points and transects in the Gulf of San Jorge; (**B**) MMI interpolated bathymetry ($\hat{f}_{CV}$); (**C**) Cross-validation local standard error $\Delta \hat{f}_{CV}$. (**D**) and (**E**) have, respectively, the same meaning but including fractal extrapolation in the algorithm.

**Table 3.** Statistics of interpolated bathymetry with real data from the Gulf of San Jorge using the MMI interpolation without and with fractal extrapolation. Values are in meters.

| $p = 0.0181$ | Simple Interpolation | Fractal Extrapolation |
|---|---|---|
| $\bar{z}$ | 81.32 | 81.24 |
| $\sigma_z$ | 24.06 | 24.20 |
| $\Delta\hat{f}_{\text{CVrms}}$ | 2.02 | 9.65 |
| $IQ_{50\%}\Delta\hat{f}_{\text{CV}}$ | 1.28 | 4.77 |
| $IQ_{90\%}\Delta\hat{f}_{\text{CV}}$ | 4.08 | 22.87 |

## 4. Discussion

Above, we presented and tested a multigrid/multiresolution interpolation (MMI) method with four good qualities: fast, with relatively low RAM requirements (in its simplest version), extensible, based on the fewest possible statistical hypotheses, and locally exact (i.e., at each pixel scale interpolated values concide with the average measured data).

### 4.1. Asessment of the Interpolations

The potential of MMI is shown in the study cases above. One (Figure 2), in an emerged topography with very different reliefs: from mountains in the north-west (nearing the Andes mountain range) to the southern plains of Patagonian steppe; in addition, a hilly structure runs almost parallel to the coast, from the city of Comodoro (at the midpoint of the GSJ) to the north which, although not having high altitudes, stands out of the sorrounding plains. The other one (Figure 3), in a submerged area combining sandy (south) and rocky (north) coasts, island chains (north), flat sedimentary bottoms (center), basin delimiting sill (east), etc. In both cases, the interpolated surface follows in the larger scales the topography, but also in the smaller ones, if enough data are available, with no appreciable oversmoothig. This is numerically shown in the close values of mean and standard deviation of the original SRTM and interpolated DEMs, with differences below 3% for the mean, and below 10% for the standard deviation and reasonable sampling density (even for transect sampling).

Regarding the interpolation using SRTM sampled data, statistical analysis shows how interpolation cross-validation errors depend strongly on both sampling density and sampling strategy (see Tables 1 and 2): random sampling gives rms standard errors ranging from about 8.5 m ($p = 0.062$) to 20 m ($p = 0.004$), while transect sampling ranges from 28 m ($p = 0.062$) to 105 m ($p = 0.004$), i.e., 4 to 5 times larger; this shows graphically the loss of accuracy far from the transects. The relationship between cross-validation $\Delta\hat{f}_{\text{CVrms}}$ and standard interpolation error $\Delta\hat{f}_{\text{rms}}$ is approximately linear in this range of sampling densities, with $\Delta\hat{f}_{\text{CVrms}}$ slightly underestimating error computed from direct comparison with the original SRTM; nevertheless $\Delta\hat{f}_{\text{CVrms}}$ lays within the 50% and 90% interquantile errors. The inclusion of fractal extrapolation adds to these errors, as expected, but not an statistically significant amount. We can draw from this in order to analyse the GSJ interpolation (Figure 3 and Table 3).

Visually, the GSJ interpolated surface does not show any marked transect artifacts. However "pimple" effects are slightly visible in that interpolation, especially at points from the POBCh data source, and especially in the northern rocky shores (which are naturally irregular) and in the flat sedimentary plateau; it is remarkable that Tukey fences did not remove these points as outliers thus these "pimples" could be just showing real bottom roughness or the need for more sampling in the voids around them. When fractal extrapolation is applied, both effects are masked to some degree by the artificial fractal roughness. A clear case is observed in front of Cape Tres Puntas, where data is scarce and yet the surface is rough, attenuating also the effect of south-leading oceanographic survey transects (but, in turn, increasing the estimated cross-validation error). On the contrary, in the western part of the Gulf the interpolated surface shows a flat bottom with and without fractal extrapolation; this agrees with the known features of the sedimentary seabed in this

zone, also confirmed by the relatively low cross-validation error in that area, although in other areas it could result from lack of data there or nearby.

### 4.2. Assessment of the Method

The idea of multigrid methods appeared in computational mathematics [71] as a way to speedup the solution of partial differential equations and has been interpreted as a preconditioner of the resulting system of linear equations. This not only makes their resolution faster but also numerically more accurate. That was also the goal of using hierarchical basis and wavelets in interpolation methods [44]. Other multiscale methods, either Laplacian/Gaussian pyramid methods in image processing [72], or other wavelet based methods have either been focused on image information representation or compression or on feature analysis [5,44]. In some sense our MMI could be related to some of them, as it uses multiple scale grids (the quadtree structure) that could be formally related to the simple Haar wavelet basis; however, it is difficult to relate those previous works with the interpolation we perform in this work, with randomly distributed point and transect samples, and mean surface estimation at each resolution.

Our MMI method is more easy to compare with other common interpolation methods such as IDW or kriging. It has in common with them that interpolated values are computed as convex linear combinations (i.e., weighted averages) of measured data, without imposing further conditions on the resulting surface. Contrary to kriging, MMI does not require computation of the semivariogram or the stationarity assumptions, which makes it, on the one hand, a (more) parameter free method and, on the other hand, more adaptable to extended areas with subareas of very different elevation profiles. Like these two methods, it is a convex method: interpolated elevations will be weighted averages of measured ones, thus it cannot predict a crest or a valley unless these features were captured by the sampling of the elevation or bathymetric surface (however, see below further improvements that can be included along these lines). MMI, whose underlying idea is just the simple spatial averaging of measuremens inside a tile, is easy to interpret, at least locally; this it has in common with OK which is the best linear unbiased predictor, that is, an estimator of the expected mean elevation based on correlated nearby measurements. The difference here is that MMI assumes measurements to be reliable and aims at interpolating the surface which contains these points, instead of the surface which is the estimated mean of the stochastic surface to which measured points belong.

However, MMI lacks the predictive capabilities of machine learning methods that can predict based on the geophysical features in the area, and are not limited to linear combinations of observations [73]. Contrary to these methods, it can only detect outliers from an statistical assessment of the interpolated bathymetry as we performed in the GSJ bathymetry, although as with any statistical assessment it is not a risk-free decission. Anyway, performing this statistical assessment of the final interpolated surface, using *K*-fold cross validation as in this work, has other advantages as the spatialization of the estimated error, which is very important in cases with inhomogeneous surfaces, as in the SRTM simulation, or inhomogeneous sampling, as in the GSJ bathymetry [74]. Other potential weakness is related with using data from different sources, but not taking into account their different levels of accuracy. In our approach we took into account these accuracy levels only regarding the vertical reference in the harmonization step (shifting each data set reference to match, on average, the previously more reliable datasets at their cross points). From there on, we applied the common method of rejecting those points far off the general surface trend [21]. Other approaches such as reweighting the data depending on their distance to the average surface (taking into account, or not, local cross-validation error), would have been against our goal of an interpolation method with the least number of assumptions.

Computationally speaking, MMI has also a number of advantages. First, interpolation time is mostly independent of the number of points as the algorithm runs on the quadtree raster pyramid; hence only final raster size determines that time (roughly mul-

tiplying it by 4 with every halving of the pixel size). This also means that it will be advantageous when interpolating a large number of data points such as in our bathymetry example: a $3346 \times 4928$ raster interpolation of 339,874 bathymetric points (padded to $4096 \times 8192$ pixels for computation) took on average 17 min on an Intel(R) Core(TM) i7-8750H CPU @ 2.20 GHz; the fractal extrapolation took longer: 120 min. Also, being based on raster local operations, it can be adapted to GPU parallel computation (something we have not addressed in our simulations). The fractal extrapolation extension, is not so time-efficient nor so easy to parallelize in the GPU; first, it involves estimating the (multi) fractal distribution parameters, and after that, it requires the use of random numbers for the simulation (which is an issue that has been addressed in other areas such as Monte Carlo simulations [75], but nevertheless increases the complexity of the GPU operations).

Our fractal extrapolation is based on the widely explored characterization of the Earth topography as multifractal [51,53,54]. It takes especially advantage of transect sampling that has been exploited in the past for fractal characterization [76]. It can be seen as a particular approach to geostatistical simulation, that attempts to include complex fine-scale features into (or onto) coarse resolution DEMs, taking into account larger scale spatial height distribution to estimate smaller ones [48]; our estimation method is parametric as it assumes a fractal model. Keeping surface roughness, even if it is simulated, helps to perform terrain classification and regionalization based on geomorphological features computed usually as focal statistics of elevation distribution [7,9], and then terrain classification based on feature distribution across the study area [39,77]; smooth interpolated areas would appear as unreal separate classes, otherwise. From the most basic interest in DEM assessment, fractal extrapolation provides a more realistic estimation of error: in areas where interpolated DEM is totally determined by distant measurements, error can be underestimated based on error propagation (assuming or not a underlying convex formula and gaussian process), or on cross-validation. However, simulating an stochastic surface with the same properties observed in measured areas, will give a more conservative error estimation. Although our method gets this, it is true that some of the simulated features are too random (due to isotropy) and do not prolong the natural trends observed in the area (see, for example, the southern area in front of Cape Tres Puntas in Figure 3).

Future improvements of the MMI algorithm may include extending the generalization window to perform a least squares approximation of the curved surface, weakening the current assumption of a locally flat surface and allowing the inclusion of anysotropy in the fractal extrapolation. This would render more realistic groove and ridge-like features in continuity with the known elevation data [51].

## 5. Conclusions

In this article, we introduced a multigrid/multiresolution interpolation (MMI) method. The goal of the method is simplicity, both in implementation and in statistical and other assumptions, and scalability to efficiently interpolate large datasets. The quadtree multigrid raster approach makes the method fast and memory efficient. This allows the use of *K*-fold cross-validation methods to compute local interpolation standard errors, which not only inform about the interpolation quality, but also, helps assess input data quality using outlier detection; this is important when working with heterogeneous data as in our Gulf of San Jorge bathymetry case study.

The (multi)fractal extrapolation method simulates natural roughness in areas with no data (e.g., between transects). On the one hand, this simulates a roughness with the same scale and statistical topographical properties observed in the data (especially in transect data) and, on the other hand, it provides a more realistic asessment of the DEM *K*-fold cross-validation uncertainty based on the well established multifractal nature of the Earth relief.

We have applied the MMI to synthetic (SRTM elevation model) and real (Gulf of San Jorge bathymetry) DEM interpolation problems, showing how errors depend on sampling strategy and density, and how *K*-fold cross-validation does a reasonably good job assessing

local and global errors. The results show visually realistic surfaces with varying levels of detail, i.e., no oversmoothing, while also reducing transect and "bump" artifacts to a minimum, across a geomorphologically rich area.

**Author Contributions:** D.R.-P. developed the mathematical and computational aspects of the article. N.S.-C. obtained the data and performed initial standardization and cleaning. Both contributed to write and revise the article. All authors have read and agreed to the published version of the manuscript.

**Funding:** This work was partially supported by the Agencia Nacional de Promoción Científica y Tecnológica (ANPCyT) of Argentina through project PICT 2016-0218.

**Institutional Review Board Statement:** Not applicable.

**Informed Consent Statement:** Not applicable.

**Data Availability Statement:** The code implementing the algorithms described and some sample data can be found in the GitHub public repository https://github.com/daniel-rperez/mrinterp (access date 17 June 2022).

**Acknowledgments:** The authors would like to thank Jesus San Martin for his suggestions about method validation. The authors would like to also acknowledge the joint project "Fortalecimiento de la Gestión y Protección de la Biodiversidad Costero Marina en Áreas Ecológicas clave y la Aplicación del Enfoque Ecosistémico de la Pesca (EEP)" between United Nations' Food and Agriculture Organization (FAO) and Ministerio de Ambiente y Desarrollo Sostenible de la Nación Argentina, GCP/ARG/025/GFF, in whose framework were tested some of the methods presented in the article.

**Conflicts of Interest:** The authors declare no conflict of interest.

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
