# Peer review of "Multigrid/Multiresolution Interpolation: Reducing Oversmoothing and Other Sampling Effects"

_2673-7418, doi:10.3390/geomatics2030014_

Round 1
Reviewer 1 Report
This paper introduces a simple but robust multigrid/multiresolution interpolation (MMI) method which adapts to the spatial resolution available, being an exact interpolator where data exist and a smoothing generalizer where data are missing. The subject of this paper is of interest to the scientific community, although significant improvements in text and methodology are required.
1.The introduction should be shorter and the content should be more purposeful, based on the objectives of the manuscript.
2.The section describing the study areas, should be added as a separate chapter before the methodology.
3. Regarding the methodology, there are significant weaknesses. The main issue is that bathymetric data were used from different sources, without taking into account the different levels of accuracy of each dataset.
4. The results should be presented in more details in a separate chapter and separated from the discussion chapter.
Author Response
Q: This paper introduces a simple but robust multigrid/multiresolution interpolation (MMI) method which adapts to the spatial resolution available, being an exact interpolator where data exist and a smoothing generalizer where data are missing. The subject of this paper is of interest to the scientific community, although significant improvements in text and methodology are required.
1.The introduction should be shorter and the content should be more purposeful, based on the objectives of the manuscript.
A: The outline of the introduction was as follows:
- Importance of (interpolated) digital elevation models
- Data sources used to generate DEMs
- The problem of DEM accuracy estimation
- Open access land DEMs
- The problem of bathymetric DEMs
- Brief review of DEM interpolation methods
- The problem of artifacts in interpolated DEMs
- DEM multiscale interpolation or superresolution
- Problems posed by those interpolation algorithms and aims of the article
We think that outline was purposeful, given the objective of our article. However, to comply with the reviewer request we have removed point 2 and abridged the introduction to 6 and the description of 7.
Q: 2.The section describing the study areas, should be added as a separate chapter before the methodology.
A: We have considered the reviewer’s suggestion, which is consistent with a strict reading of the journal guidelines. However, we have kept our sectioning with a description of the methodology and the study cases, both preceded by the description of the study area, in order to make the article easier to read having the relevant information about the study area closer to where our interpolation results are described.
Q: 3. Regarding the methodology, there are significant weaknesses. The main issue is that bathymetric data were used from different sources, without taking into account the different levels of accuracy of each dataset.
A: The reviewer is right, we had not emphasized that issue, common in crowdsourcing methodologies that use data provided by users with different reliabilities. However, there is a good reason for that: our data, acquired in the context of scientific campaigns, had already been curated (and we rechecked their spatial consistency when building the database). We have added a remark clarifying this QA. We have also included in our discussion this point raised by the reviewer.
Q: 4. The results should be presented in more details in a separate chapter and separated from the discussion chapter.
A: The results of the study cases were already separated from the discussion section, as the reviewer requests. We understand that all the relevant results are presented with detail in tables 1-3 and figures 2-3, and it would be repetitive just to describe them again in a new section or subsection. On the other hand, these results are discussed in detail, both for the study cases, as well as from the point of view of the strengths and weaknesses of the proposed interpolation method.
Reviewer 2 Report
Section 2.2
What does "exact interpolator" mean? Interpolation assumes some prior knowledge of function behavior. What is assumed in the approach suggested by the authors? If this means that the interpolated surface includes known data points, this may not be the optimal interpolation as the acquired data always include measurement errors.
Specific comments:
Line 138: "hypotheses"
Line 144,402: probably "outlier", not "outlayer"
Line 159: "...a priori knowledge"?
Line 176: what are x_k? Known function values? Please state explicitly.
Line 262,263: "an" -> "a"
Line 277: "based in" -> "based on"
Figure 2: standard error coloring is not clear
Line 373: "nautic" -> "nautical"
Line 403: "Tucker" -> "Tukey"
Author Response
Q: What does "exact interpolator" mean?
A: We use “exact interpolator” with its usual meaning, that is, that the interpolated function takes at the coordinates of measurement points the measured values. Most interpolation methods are exact, although some make room for non-exactness as in the case of kriging, if an intrinsic surface roughness is considered (the semivariogram “nugget”).
In the case of our algorithm, we generalize this definition of exactness to average measured values within each resolution pixels:
Exactness: The method is an exact interpolator meaning that, for any partition of I in 2n x 2n subintervals, the interpolated fÌ‚(Bij) is the mean of observed values of f at points within Bij ⊂ I, in particular for Bij containing one single point (that is the usual meaning of exact interpolation method).
Q: Interpolation assumes some prior knowledge of function behavior. What is assumed in the approach suggested by the authors?
A: The only assumption is local scale continuity, which is behind our approach of interpolating no-data cells with neighborhood averages at each scale. This is the simplest hypothesis we can think of at each resolution, nevertheless its natural generalization to finer resolutions renders a final smooth surface (unless fractal extrapolation is applied).
Q: If this means that the interpolated surface includes known data points, this may not be the optimal interpolation as the acquired data always include measurement errors.
A: This is a problem with any interpolation method: how does the method judge which point is an outlier and which is just a local surface deviation (e.g. a ditch)? Our interpolation method will not solve that problem, but our k-fold cross-validation could help detect (and discard, through Tukey fences) those outliers and render a better approximation (the k-fold average) of the interpolated surface, as well as a local error estimation.
Q: Specific comments:
Line 138: "hypotheses"
Line 144,402: probably "outlier", not "outlayer"
Line 159: "...a priori knowledge"?
Line 176: what are x_k? Known function values? Please state explicitly.
Line 262,263: "an" -> "a"
Line 277: "based in" -> "based on"
Line 373: "nautic" -> "nautical"
Line 403: "Tucker" -> "Tukey"
A: We have corrected all these typos. Thank you so much for your detailed specification.
Q: Figure 2: standard error coloring is not clear
A: We have improved the figure quantiles colorbar, solving a problem in our R plotting code.
Reviewer 3 Report
The manuscript introduced a robust multigrid/multiresolution interpolation (MMI) method. The topic, presentation, structure are correct. A case study was provided to test the proposed method. I accept this manuscript in its present form.
Author Response
Q: The manuscript introduced a robust multigrid/multiresolution interpolation (MMI) method. The topic, presentation, structure are correct. A case study was provided to test the proposed method. I accept this manuscript in its present form.
A: We thank the reviewer for his/her support and appreciation of our work.
Reviewer 4 Report
Brief report for the manuscript geomatics-1765256.
For the detailed report, please refer to the attached PDF file.
The paper presents the general study goal: to find the optimal interpolation method, which would be better compared to conventional methods. The main goal was to compute bathymetric data and interpolate DEM in the study area of Gulf of San Jorge (Argentina) using new method using DEM for data.
The authors identified research gap: existing methods of interpolation, such as IDW, kriging, radial basis functions, regularized splines, are not suitable for the non-homogeneous data where they show over/under-smoothing of the interpolated surface depending on local point density. This leads to the loss of actual information in high point density areas or generation of artifacts.
Therefore, they presented a robust multigrid/multi-resolution interpolation (MMI) method which adapts to the spatial resolution of spatial data. Using their novel method the authors generated bathymetric surfaces in the coastal regio nof Argentic with estimated local validation errors using DEM, within the 10% of the bathymetric surface typical deviation in the real calculation.
The submission adheres to the Geomatics journal policies and general standards. The article conforms to general professional standards of GIS/geosciences domains by expressions used in the text. The article includes sufficient introduction background to demonstrate how the work fits into the broader field of knowledge in topography and what research gaps are covered.
Figures and maps are relevant to the content of the article. They are of sufficient resolution, appropriately described and labeled using continuous numeration.
The present results are compared with other studies in the discussion. The advantages/disadvantages of the proposed method with respect to the obtained results are described. The novelty of present paper compared to the existing studies is described which justified that it deserves to be published in the Geomatics journal.
This manuscript can be accepted based on the detailed report above.
I only suggest some small minor corrections colored yellow in the attached PDF file.
With kind regards, - Reviewer.
27.05.2022.

Author Response
Q: The paper presents the general study goal: to find the optimal interpolation method, which would be better compared to conventional methods. The main goal was to compute bathymetric data and interpolate DEM in the study area of Gulf of San Jorge (Argentina) using a new method using DEM for data.
The authors identified research gap: existing methods of interpolation, such as IDW, kriging, radial basis functions, regularized splines, are not suitable for the non-homogeneous data where they show over/under-smoothing of the interpolated surface depending on local point density. This leads to the loss of actual information in high point density areas or generation of artifacts.
Therefore, they presented a robust multigrid/multi-resolution interpolation (MMI) method which adapts to the spatial resolution of spatial data. Using their novel method the authors generated bathymetric surfaces in the coastal region of Argentina with estimated local validation errors using DEM, within the 10% of the bathymetric surface typical deviation in the real calculation.
The submission adheres to the Geomatics journal policies and general standards. The article conforms to general professional standards of GIS/geosciences domains by expressions used in the text. The article includes sufficient introduction background to demonstrate how the work fits into the broader field of knowledge in topography and what research gaps are covered.
Figures and maps are relevant to the content of the article. They are of sufficient resolution, appropriately described and labeled using continuous numeration.
The present results are compared with other studies in the discussion. The advantages/disadvantages of the proposed method with respect to the obtained results are described. The novelty of present paper compared to the existing studies is described which justified that it deserves to be published in the Geomatics journal.
This manuscript can be accepted based on the detailed report above.
A: We thank the reviewer so much for his/her thorough revision of our manuscript.
Q: Keywords. I suggest to add the following ones to better highlight the study areas:
Patagonia, Argentina, Atlantic Ocean.
A: We have added these keywords to help indexation by study area.
Q: Main Text. Introduction. Correct: prediction, . . . are just a few of their current applications [line 26].
A: We have corrected that text following the reviewer's suggestion.
Q: Main Text. 3.3. Gulf of San Jorge bathymetry interpolation. Correct: “Data from nautical charts” [line 373].
A: We have corrected the spelling.
Round 2
Reviewer 1 Report
The authors have answered sufficiently all my questions.